# Replication Study: Wnt activity defines colon cancer stem cells and is regulated by the microenvironment

**Anthony Essex[1], Javier Pineda[1], Grishma Acharya[2], Hong Xin[2], James Evans[1], Reproducibility Project: Cancer Biology***

[1]PhenoVista Biosciences, San Diego, United States; [2]Explora BioLabs Inc, San Diego, United States

**\*For correspondence:**
tim@cos.io;
nicole@scienceexchange.com

**Group author details:**
Reproducibility Project: Cancer Biology See page 14

**Abstract** As part of the Reproducibility Project: Cancer Biology we published a Registered Report (Evans et al., 2015), that described how we intended to replicate selected experiments from the paper 'Wnt activity defines colon cancer stem cells and is regulated by the microenvironment' (Vermeulen et al., 2010). Here, we report the results. Using three independent primary spheroidal colon cancer cultures that expressed a Wnt reporter construct we observed high Wnt activity was associated with the cell surface markers CD133, CD166, and CD29, but not CD24 and CD44, while the original study found all five markers were correlated with high Wnt activity (Figure 2F; Vermeulen et al., 2010). Clonogenicity was highest in cells with high Wnt activity and clonogenic potential of cells with low Wnt activity were increased by myofibroblast-secreted factors, including HGF. While the effects were in the same direction as the original study (Figure 6D; Vermeulen et al., 2010) whether statistical significance was reached among the different conditions varied. When tested *in vivo*, we did not find a difference in tumorigenicity between high and low Wnt activity, while the original study found cells with high Wnt activity were more effective in inducing tumors (Figure 7E; Vermeulen et al., 2010). Tumorigenicity, however, was increased with myofibroblast-secreted factors, which was in the same direction as the original study (Figure 7E; Vermeulen et al., 2010), but not statistically significant. Finally, we report meta-analyses for each results where possible.
DOI: https://doi.org/10.7554/eLife.45426.001

## Introduction

The Reproducibility Project: Cancer Biology (RP:CB) is a collaboration between the Center for Open Science and Science Exchange that seeks to address concerns about reproducibility in scientific research by conducting replications of selected experiments from a number of high-profile papers in the field of cancer biology (*Errington et al., 2014*). For each of these papers, a Registered Report detailing the proposed experimental designs and protocols for the replications was peer reviewed and published prior to data collection. The present paper is a Replication Study that reports the results of the replication experiments detailed in the Registered Report (*Evans et al., 2015*) for a paper by *Vermeulen et al. (2010)* and uses a number of approaches to compare the outcomes of the original experiments and the replications.

*Vermeulen et al. (2010)* reported that colon cancer cell subpopulations with high Wnt activity correlated with markers of cancer stems cells (CSC) and displayed enhanced tumor initiating potential. Moreover, factors secreted from cancer-associated fibroblasts (CAFs), which are an important component of the stroma, such as HGF, were reported to play a role in the formation of the CSC niche and tumorigenicity by activating the Wnt signaling pathway (*Vermeulen et al., 2010*). This was

demonstrated using *in vitro* clonogenicity and *in vivo* tumorigenicity assays, suggesting Wnt activity defines CSCs and is regulated by the microenvironment.

The Registered Report for the paper by *Vermeulen et al. (2010)* described the experiments to be replicated (Figures 2F, 6D and 7E), and summarized the current evidence for these findings (*Evans et al., 2015*). Since that publication additional studies have reported a relationship between Wnt activity, using a Wnt reporter like *Vermeulen et al. (2010)*, and CSC properties in various malignancies, including colorectal, lung, gastric, and breast cancer (*Jun et al., 2016*; *Su et al., 2015*). Moreover, recent studies have also reported CSC properties from cells with high expression of Wnt target genes, such as *LGR5* (*Dame et al., 2018*; *Junttila et al., 2015*; *Shimokawa et al., 2017*). Furthermore, recent studies have continued to examine the role of the microenvironment and cancer stemness. Niche factor requirements in colorectal tumors were found to decrease during tumorigenesis (*Fujii et al., 2016*; *Kashfi et al., 2018*). While a new modeling approach suggested stem cell functionality during colorectal tumor expansion was defined by secreted factors from CAFs rather than cell-intrinsic properties (*Flanagan et al., 2018*; *Lenos et al., 2018*).

*Vermeulen et al. (2010)* also reported CD133, the combination of CD29/CD24, and the combination of CD44/CD166 were correlated with high Wnt activity. CD133 has been suggested to mark CSCs in various tumor types, although the accuracy as a CSC biomarker has been highly controversial (*Glumac and LeBeau, 2018*). In colorectal cancer, variation in clonogenic potential with specific cell populations have been reported (*LaBarge and Bissell, 2008*), with CD133$^+$ cells reported to be associated with the CSC population in two separate studies (*O'Brien et al., 2007*; *Ricci-Vitiani et al., 2007*), while *Shmelkov et al. (2008)* reported both CD133$^+$ and CD133$^-$ populations were capable of forming colonospheres *in vitro* and were serially tumorigenic in mice. Variation has also been reported in independent studies that examined CD133 expression to define a clonogenic subfraction when examining the same cell line, HCT116 (*Chen et al., 2011*; *Dittfeld et al., 2009*). As such, other studies have reported other markers, such as CD44 and CD166, to be more robust in identifying colorectal CSCs (*Dalerba et al., 2007*; *Ozawa et al., 2014*). There is also variation of the significance of CD24 expression in colorectal cancer, while the significance of CD29 needs further investigation, although the presence of these molecules have been associated with CSC characteristics (*Hatano et al., 2017*; *Izumi et al., 2015*; *Muraro et al., 2012*).

The outcome measures reported in this Replication Study will be aggregated with those from the other Replication Studies to create a dataset that will be examined to provide evidence about reproducibility of cancer biology research, and to identify factors that influence reproducibility more generally.

## Results and discussion

### Generation and characterization of primary spheroidal cultures of colon cancer cells

To assess Wnt signaling activity in colon cancer stem cells (CSC), we transduced primary spheroidal cultures of colon cancer cells with the same Wnt reporter construct as the original study, which used a TCF/LEF-1 responsive promoter to drive expression of green fluorescent protein (GFP) (TOP-GFP; *Reya et al., 2003*). The experimental approach to generate TOP-GFP expressing CSC cultures was described in Protocol 1 of the Registered Report (*Evans et al., 2015*). We used three independent spheroidal cultures, one used in the original study (Co100) and two derived from primary human colorectal cancer tissues (CSC1 and E450). The three cultures were transduced with TOP-GFP and single-cell TOP-GFP cultures were isolated. This approach, similar to the original study, was done to exclude variation in lentiviral integration and copy number between cells (*Vermeulen et al., 2008*). The single-cell-derived TOP-GFP cultures displayed heterogeneity in Wnt signaling, similar to what was reported in the original study (*Figure 1A,B*).

The three TOP-GFP cultures were then sorted into the highest and lowest 10% of TOP-GFP-expressing cells by fluorescence-activated cell sorting and analyzed for expression of the same cell surface markers reported in the original study. This experiment is similar to what was reported in *Figure 2F* of *Vermeulen et al. (2010)* and described in Protocol 2 in the Registered Report

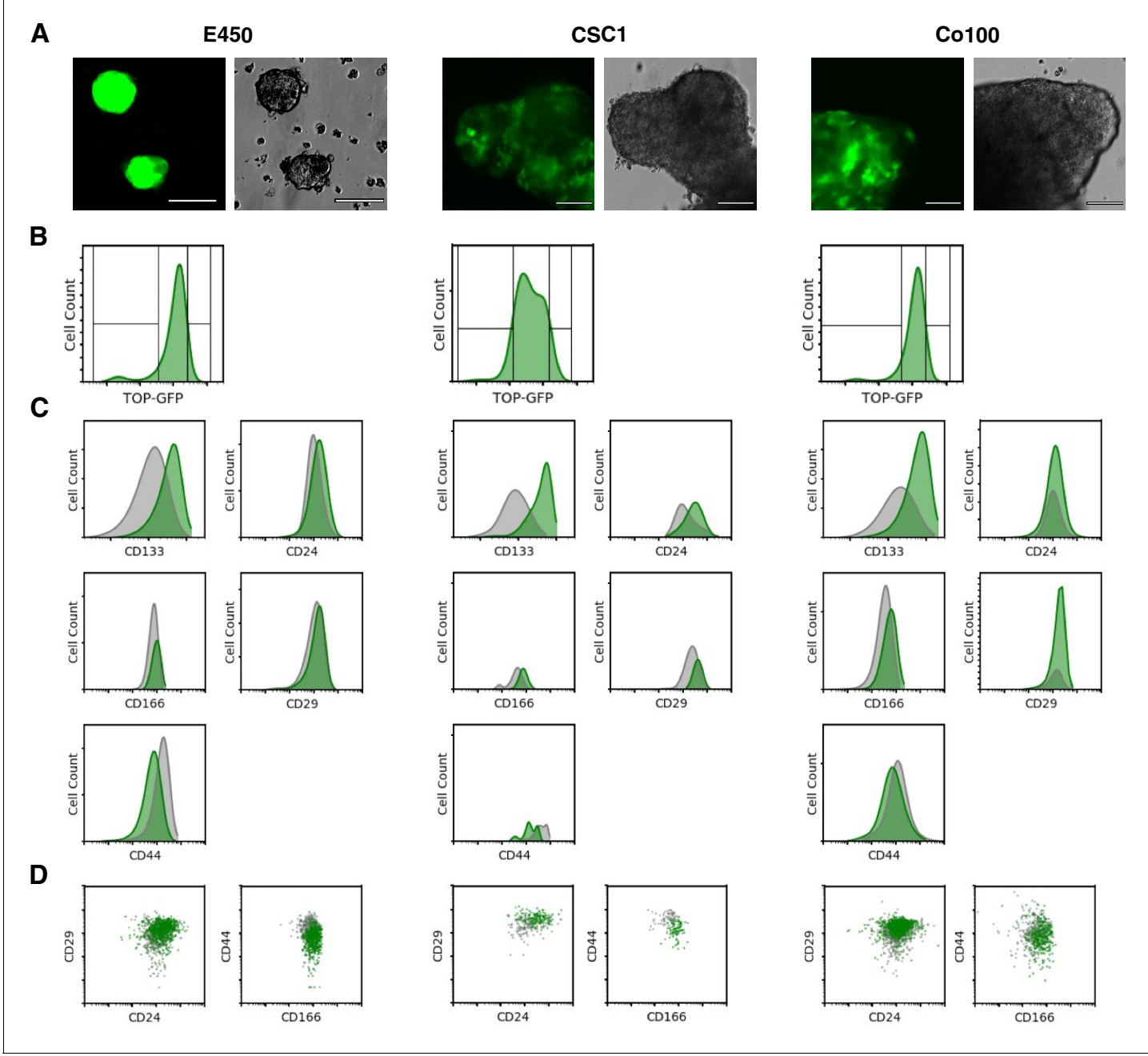

**Figure 1.** Analysis of CSC marker expression in TOP-GFP cultures. (A) Representative images of the three independent single-cell-cloned CSC cultures, lentivirally transduced with TOP-GFP. Phase contrast (top) and fluorescence microscopy (bottom) for each of the cultures indicated. Bar = 90 μm. (B) Single parameter histograms for GFP intensity for each of the TOP-GFP single-cell-cloned CSC cultures with the TOP-GFP$^{low}$ (10% lowest) and TOP-GFP$^{high}$ (10% highest) populations indicated. (C) Single parameter histograms for the indicated cell surface markers for each of the indicated cultures. Gray denotes TOP-GFP$^{low}$ (10% lowest) and green denotes TOP-GFP$^{high}$ (10% highest) populations. (D) Density plots for CD29/CD24 and CD44/CD166 from TOP-GFP$^{low}$ (gray) and TOP-GFP$^{high}$ (green) populations of each culture. Additional details for this experiment can be found at https://osf.io/tfy28/ .

DOI: https://doi.org/10.7554/eLife.45426.002

The following figure supplement is available for figure 1:

**Figure supplement 1.** Flow cytometry gating strategy.

DOI: https://doi.org/10.7554/eLife.45426.003

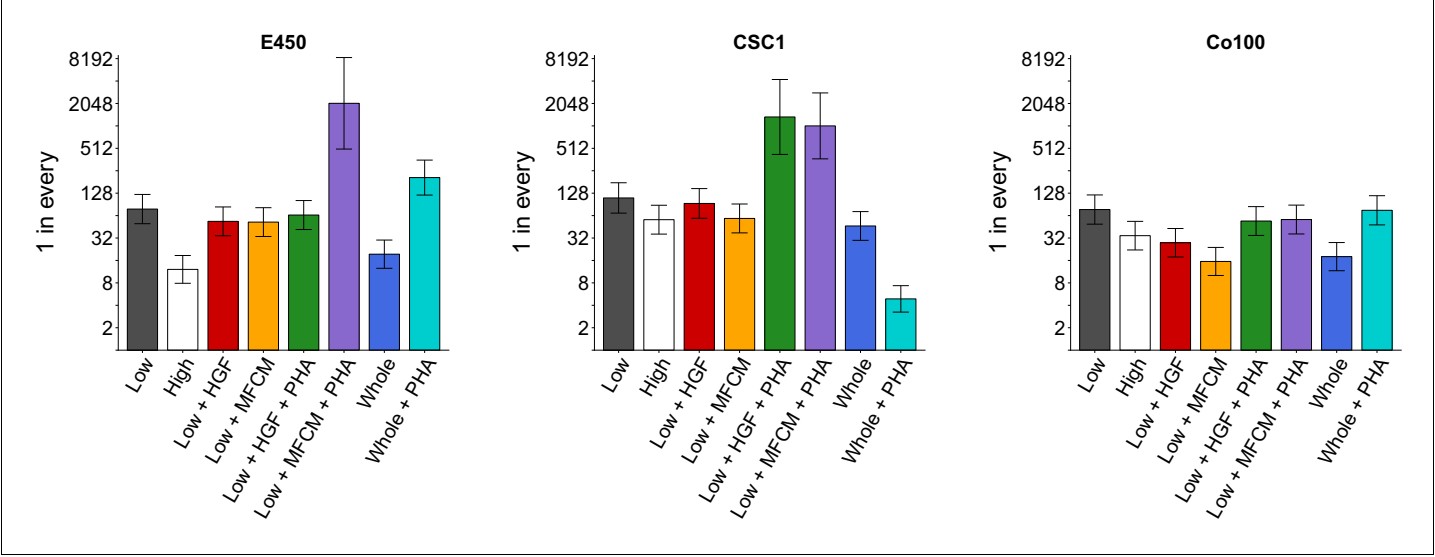

**Figure 2.** Clonogenicity assay of TOP-GFP cultures. A limiting-dilution assay was performed on the TOP-GFP[low], TOP-GFP[high], or TOP-GFP[whole] populations of the three indicated TOP-GFP cultures. Cells were left untreated, or treated with 25 ng/ml HGF, 1:2 dilution of MFCM, or 500 nM PHA-665752 (PHA), as indicated. The bar graphs present the clonogenic potential of each culture with error bars representing 95% confidence intervals (y-axis is $\log_2$ scale). This experiment was performed once for each culture. See Materials and methods and Registered Report (*Evans et al., 2015*) for details on limiting-dilution statistics and scheme. Planned contrast between TOP-GFP[low] vs TOP-GFP[high]: E450 ($\chi^2 = 39.8$, uncorrected $p=2.82\times10^{-10}$, corrected $p=1.69\times10^{-9}$); CSC1 ($\chi^2 = 4.82$, uncorrected $p=0.028$, corrected $p=0.169$); Co100 ($\chi^2 = 7.59$, uncorrected $p=0.0059$, corrected $p=0.035$). Planned contrast between TOP-GFP[low] vs TOP-GFP[low] + HGF: E450 ($\chi^2 = 1.49$, uncorrected $p=0.223$, corrected $p>0.99$); CSC1 ($\chi^2 = 0.337$, uncorrected $p=0.562$, corrected $p=0.99$); Co100 ($\chi^2 = 12.7$, uncorrected $p=3.70\times10^{-4}$, corrected $p=0.0022$). Planned contrast between TOP-GFP[low] vs TOP-GFP[low] + MFCM: E450 ($\chi^2 = 1.96$, uncorrected $p=0.162$, corrected $p=0.969$); CSC1 ($\chi^2 = 4.18$, uncorrected $p=0.041$, corrected $p=0.245$); Co100 ($\chi^2 = 28.7$, uncorrected $p=8.26\times10^{-8}$, corrected $p=4.96\times10^{-7}$). Planned contrast between TOP-GFP[low] + HGF vs TOP-GFP[low] + HGF + PHA: E450 ($\chi^2 = 0.376$, uncorrected $p=0.540$, corrected $p>0.99$); CSC1 ($\chi^2 = 34.0$, uncorrected $p=5.64\times10^{-9}$, corrected $p=3.39\times10^{-8}$); Co100 ($\chi^2 = 5.13$, uncorrected $p=0.024$, corrected $p=0.141$). Planned contrast between TOP-GFP[low] + MFCM vs TOP-GFP[low] + MFCM + PHA: E450 ($\chi^2 = 61.0$, uncorrected $p=5.71\times10^{-15}$, corrected $p=3.43\times10^{-14}$); CSC1 ($\chi^2 = 43.5$, uncorrected $p=4.14\times10^{-11}$, corrected $p=2.48\times10^{-10}$); Co100 ($\chi^2 = 17.6$, uncorrected $p=2.67\times10^{-5}$, corrected $p=1.60\times10^{-4}$). Planned contrast between TOP-GFP[whole] vs TOP-GFP[whole] + PHA: E450 ($\chi^2 = 68.3$, uncorrected $p=1.43\times10^{-16}$, corrected $p=8.56\times10^{-16}$); CSC1 ($\chi^2 = 72.2$, uncorrected $p=1.96\times10^{-17}$, corrected $p=1.17\times10^{-16}$); Co100 ($\chi^2 = 20.2$, uncorrected $p=6.91\times10^{-6}$, corrected $p=4.14\times10^{-5}$). Additional details for this experiment can be found at https://osf.io/k9vce/.
DOI: https://doi.org/10.7554/eLife.45426.004

The following figure supplement is available for figure 2:

**Figure supplement 1.** Pilot of clonogenicity assay.
DOI: https://doi.org/10.7554/eLife.45426.005

(*Evans et al., 2015*). We found the TOP-GFP[high] populations were more enriched for CD133[+] or CD166[+] cells compared to the TOP-GFP[low] populations for each of the three cultures (*Figure 1C,D*). There were also more CD29[+] cells in the TOP-GFP[high] populations for each of the three cultures, while the two TOP-GFP populations were mostly similar for CD24 expression. We also found the TOP-GFP[low] populations from E450 and CSC1 cultures were more enriched for CD44[+] cells, while both the populations displayed similar expression for Co100. The original study stated that CD133, the combination of CD29/CD24, and the combination of CD44/CD166 were correlated with the TOP-GFP[high] population (*Vermeulen et al., 2010*). However, since the degree that each of the markers correlated with the TOP-GFP[high] and TOP-GFP[low] populations were not completely reported in the original study, it is difficult to directly compare to the results reported in this replication attempt. To summarize, for this experiment we found results that varied in direction relative to the original study.

## Clonogenicity of TOP-GFP CSC cultures

The three TOP-GFP cultures were then used to assess the clonogenic potential of the cells using a limiting-dilution assay. Different TOP-GFP expressing fractions were examined to test if variation in TOP-GFP levels resulted in differential clonogenicity. Additionally, TOP-GFP[low] fractions were treated with conditioned medium derived from myofibroblasts (MFCM) or hepatocyte growth factor (HGF), with and without a specific c-Met inhibitor (PHA665752), to test if myofibroblast-secreted factors increased the clonogenic potential. This experiment is similar to what was reported in Figure 6D of *Vermeulen et al. (2010)* and described in Protocol 3 in the Registered Report (*Evans et al., 2015*). We first performed a pilot assay of the dilution curve in untreated conditions and observed the clonogenic potential in the TOP-GFP[high] fractions were greater than the TOP-GFP[low] fractions for each of the cultures tested (*Figure 2—figure supplement 1*). We then proceeded with the experiment to test all of the conditions specified in the Registered Report and reported in the original study. Similar to the pilot assay, we found the clonogenic potential of TOP-GFP[high] cells were greater than TOP-GFP[low] cells for each of the three cultures (*Figure 2*). We also observed that the clonogenicity of TOP-GFP[low] cells were increased in the presence of MCFM or HGF, which was reduced when PHA665752 was included, although to varying degrees across the different cultures. Interestingly, PHA665752 treatment on the whole population of TOP-GFP cells (TOP-GFP[whole]) had varying effects on the clonogenicity among the different cultures tested. Both the Co100 and E450 cultures had decreased clonogenicity in the presence of PHA665752, while CSC1 cultures were increased. The original study reported the clonogenic potential of the TOP-GFP[high] fraction was greater compared to TOP-GFP[low] cells, with the clonogenic potential of TOP-GFP[low] cells enhanced with MFCM, or HGF, treatment, almost to the level of TOP-GFP[high] cells, which was blocked with PHA665752 (*Vermeulen et al., 2010*). PHA665752 was also reported to have no effect on the clonogenicity of TOP-GFP[whole] cells (*Vermeulen et al., 2010*). The HGF and PHA665752 concentrations were the same between the original study and this replication attempt (25 ng/ml and 500 nM, respectively) as was the MFCM treatment that used a 1:2 dilution of MFCM diluted in CSC medium. As suggested during peer review of the Registered Report (*Gilbertson, 2015*), we also determined the concentration of HGF in MFCM, which was determined by enzyme-linked immunosorbent assay (ELISA) to be 0.61 ng/ml. The original study reported HGF production in MFCM was ~120 ng/ml (Figure 5E; *Vermeulen et al., 2010*) or approximately 200 times higher than what we observed. Other studies that measured the amount of HGF in MFCM using the same cell line (18Co) and timeline (24 hr) reported concentrations of ~0.4 ng/ml (*Shao et al., 2006*) and ~6 ng/ml (*Woo et al., 2015*). The variation of HGF production might be explained by differences in assay reagents, such as the generation of the standard curve (*Jones et al., 1995*) and variability in microplate surface properties (*Lilyanna et al., 2018*). Further, HGF production has been shown to be influenced by other soluble factors, such as transforming growth factor-β (TGF-β) and basic fibroblast growth factor (FGF2) (*Neuss et al., 2004*). The variation in HGF production in MFCM between the original study and this replication attempt might account for any observed differences in outcomes and should be taken into account when interpreting these results. Importantly, though, observing and reporting all outcomes are informative to establish the range of conditions under which a given phenotype can be observed (*Bailoo et al., 2014*).

As outlined in the Registered Report (*Evans et al., 2015*), we planned to conduct six comparisons using the Bonferroni correction to adjust for multiple comparisons, making the *a priori* significance threshold 0.0083. We performed Extreme Limiting Dilution Analysis (ELDA) (*Hu and Smyth, 2009*) and tested for pairwise differences in frequency between groups (see *Figure 2* figure legend). The sample sizes were determined *a priori* to detect the effects based on the originally reported data. We found that the test between TOP-GFP[high] and TOP-GFP[low] cells was statistically significant for Co100 and E450 cultures, but not CSC1. Treatment with HGF, or MFCM, resulted in a statistically significant increase in clonogenicity in TOP-GFP[low] cells from the Co100 culture, but not E450 or CSC1 cultures. The comparison of HGF treatment, with or without PHA665752, in TOP-GFP[low] cells was statistically significant for the CSC1 culture, but not the Co100 or E450 cultures, while the comparison of MFCM treatment, with or without PHA665752, in TOP-GFP[low] cells was statistically significant for all three cultures. Furthermore, the differences observed in TOP-GFP[whole] cells with or without PHA665752 were statistically significant for all three cultures. To summarize, for this experiment we found results that were in the same direction as the original study, except for treatment of

TOP-GFP[whole] cells with PHA665752, and statistical significance that varied among the three cultures tested as well as the original study.

## Tumorigenicity of TOP-GFP CSC culture

We also examined the frequency TOP-GFP cells form tumors when injected into nude mice. This experiment is similar to what was reported in Figure 7E of *Vermeulen et al. (2010)* and described in Protocol 4 in the Registered Report (*Evans et al., 2015*). While the original study included TOP-GFP[low] cells co-injected with myofibroblasts and TOP-GFP[whole] cells, this replication attempt was restricted to TOP-GFP[high], TOP-GFP[low], and TOP-GFP[low] cells co-injected with MFCM. The original study also reported results from two clones, while this replication attempt utilized a single clone. As stated in the Registered Report, we identified the clone to use as the one with the largest observed difference in clonogenicity between untreated TOP-GFP[high] and TOP-GFP[low] cells, which, as described above, was the E450 culture. Different cell numbers were injected into female nude mice and blindly analyzed for tumor formation after nine weeks. We found the frequency of tumorigenicity was similar when TOP-GFP[high] cells (1 in every 3332, 95% CI [9174, 1210]) or TOP-GFP[low] cells (1 in every 2744, 95% CI [7377, 1020]) were injected (*Table 1*), which was not a statistically significant difference ($\chi^2 = 0.084$, uncorrected $p=0.772$, corrected $p>0.99$). The addition of MFCM to TOP-GFP[low] cells resulted in an increased frequency of tumorigenicity (1 in every 774, 95% CI [2268, 264]), which was not statistically significant when compared to untreated TOP-GFP[low] cells ($\chi^2 = 3.32$, uncorrected $p=0.069$, corrected $p=0.137$). The original study reported for each of the two clones tested (C100.B5 and C100.G7) the TOP-GFP[high] fraction was more effective in inducing tumors (C100.B5:~1 in every 37, 95% CI [92, 15]; C100.G7:~1 in every 961, 95% CI [2498, 369]) than the TOP-GFP[low] fraction (C100.B5:~1 in every 6939, 95% CI [18841, 2555]; C100.G7: frequency estimate unable to be determined) and that tumorigenicity was increased when TOP-GFP[low] cells were co-injected with MFCM (C100.B5:~1 in every 310, 95% CI [789, 122]; C100.G7:~1 in every 2352, 95% CI [5236, 1056]) (*Vermeulen et al., 2010*). To summarize, for this experiment we found results that were in the same direction as the original study for the comparison between TOP-GFP[low] with or without MFCM, but not for the comparison of TOP-GFP[low] and TOP-GFP[high], and not statistically significant where predicted.

## Meta-analysis of original and replication effects

We performed a meta-analysis using a random-effects model, where possible, to combine each of the effects described above as pre-specified in the confirmatory analysis plan (*Evans et al., 2015*). To provide a standardized measure of the effect, a common effect size was calculated for each effect from the original and replication studies. Cohen's $\omega$ is a standardized measure of the association between two variables, in this case the cells tested and clonogenic, or tumorigenic, frequency. The estimate of the effect size of one study, as well as the associated uncertainty (i.e. confidence interval), compared to the effect size of the other study provides another approach to compare the original and replication results (*Errington et al., 2014*; *Valentine et al., 2011*). Importantly, the width of

**Table 1.** Tumorigenicity assay of TOP-GFP culture.

Cell numbers from the indicated populations were injected into female athymic nude mice. Cells were left untreated or treated with 1:2 dilution of MFCM for 2 hr before injection. The number of successful tumor initiations after nine weeks out of four injected mice for each condition is reported. Planned contrast between TOP-GFP[low] vs TOP-GFP[high] ($\chi^2 = 0.084$, uncorrected $p=0.772$, corrected $p>0.99$). Planned contrast between TOP-GFP[low] vs TOP-GFP[low] + MFCM ($\chi^2 = 3.32$, uncorrected $p=0.069$, corrected $p=0.137$). Additional details for this experiment can be found at https://osf.io/j73xu/.

| Line | Condition | 10 | 100 | 1000 | 5000 |
|------|-----------|-----|------|------|------|
| E450 | TOP-GFP Low | 0/4 | 0/4 | 2/4 | 3/4 |
| | TOP-GFP High | 0/4 | 0/4 | 3/4 | 2/4 |
| | TOP-GFP Low + MFCM | 0/4 | 2/4 | 2/4 | 4/4 |

DOI: https://doi.org/10.7554/eLife.45426.006

the confidence interval for each study is a reflection of not only the confidence level (e.g. 95%), but also variability of the sample (e.g. *SD*) and sample size.

There were six comparisons of the *in vitro* clonogenicity assay, which were reported in *Figure 2* of this study and Figure 6D of *Vermeulen et al. (2010)*. The effect size point estimates of the original study for each of the effects was not within the 95% CI of the replication results, and vice versa (*Figure 3A*). Furthermore, the effect sizes were larger in the original study, compared to the three TOP-GFP cultures tested in this replication attempt, with the exception of treatment of TOP-GFP^{whole} cells with or without PHA665752. The meta-analyses were statistically significant for the comparison of untreated TOP-GFP^{low} and MFCM-treated TOP-GFP^{low} (*p*=0.047), but not for the other five comparisons (see *Figure 3* figure legend). Additionally, for the comparison of untreated TOP-GFP^{low} and untreated TOP-GFP^{high} as well as untreated TOP-GFP^{low} and HGF treated TOP-GFP^{high}, the large CI of the meta-analyses along with statistically significant Cochran's *Q* tests (*p*=0.0085 and *p*=0.0028, respectively) suggest heterogeneity between the original and replication studies.

There were two comparisons of the *in vivo* tumorigenicity assay, which were reported in *Table 1* of this study and Figure 7E of *Vermeulen et al. (2010)*. Similar to the clonogenicity assay, the effect sizes were larger in the original study compared to this replication attempt, and the point estimates of each study were not within the 95% CI of the other study (*Figure 3B*). The meta-analysis of the TOP-GFP^{low} and TOP-GFP^{high} comparison was not statistically significant (*p*=0.330) with a large 95% CI and a statistically significant Cochran's *Q* test (*p*=$1.35 \times 10^{-55}$) that suggests heterogeneity between the original and replication studies. The meta-analysis of untreated TOP-GFP^{low} and MFCM treated TOP-GFP^{low} was statistically significant (*p*=0.033), suggesting the null hypothesis that MFCM treatment does not impact tumorigenicity of TOP-GFP^{low} cells can be rejected; however, the large 95% CI and a statistically significant Cochran's *Q* test (*p*=0.011) suggest heterogeneity between the original and replication studies.

This direct replication provides an opportunity to understand the present evidence of these effects. Any known differences, including reagents and protocol differences, were identified prior to conducting the experimental work and described in the Registered Report (*Evans et al., 2015*). However, this is limited to what was obtainable from the original paper and through communication with the original authors, which means there might be particular features of the original experimental protocol that could be critical, but unidentified. So while some aspects, such as the TOP-GFP reporter plasmid, cell surface markers, treatment conditions of cultures, Co100 culture, and mouse strain were maintained, others were unknown or not easily controlled for. These include variables such as cell line genetic drift (*Hughes et al., 2007*; *Kleensang et al., 2016*), genetic heterogeneity of mouse inbred strains (*Casellas, 2011*), the microbiome of recipient mice (*Macpherson and McCoy, 2015*), housing temperature in mouse facilities (*Kokolus et al., 2013*), lot variability of key reagents such as HGF and PHA-665752 (*Leek et al., 2010*), and similarities and differences in patient characteristics (*Klevorn and Teague, 2016*). Environmental differences such as husbandry staff, bedding type and source, light levels, and other intangibles, all of which, by necessity, differed between the studies also affect experimental outcomes with mice (*Howard, 2002*; *Jensen and Ritskes-Hoitinga, 2007*; *Nevalainen, 2014*; *Sorge et al., 2014*). The difference in HGF production in conditioned medium between the original study and this replication attempt, as described above, is another factor to consider. Also, differences in CSC features between the cultures used in this replication attempt, as well as the original study, is another important factor to consider. This includes the expression of the cell surface markers between the TOP-GFP^{low} and TOP-GFP^{high} populations, particularly CD24 and CD44, which have been reported as Wnt target genes (*Shulewitz et al., 2006*; *Wielenga et al., 1999*). This could be due to differences in lentiviral integration and copy number of the TOP-GFP reporter, clonal artifacts, and genetic differences between the cancer cells the cultures were derived from as well as genetic drift during passaging of the cultures (*Ben-David et al., 2018*). Whether these or other factors influence the outcomes of this study is open to hypothesizing and further investigation, which is facilitated by direct replications and transparent reporting.

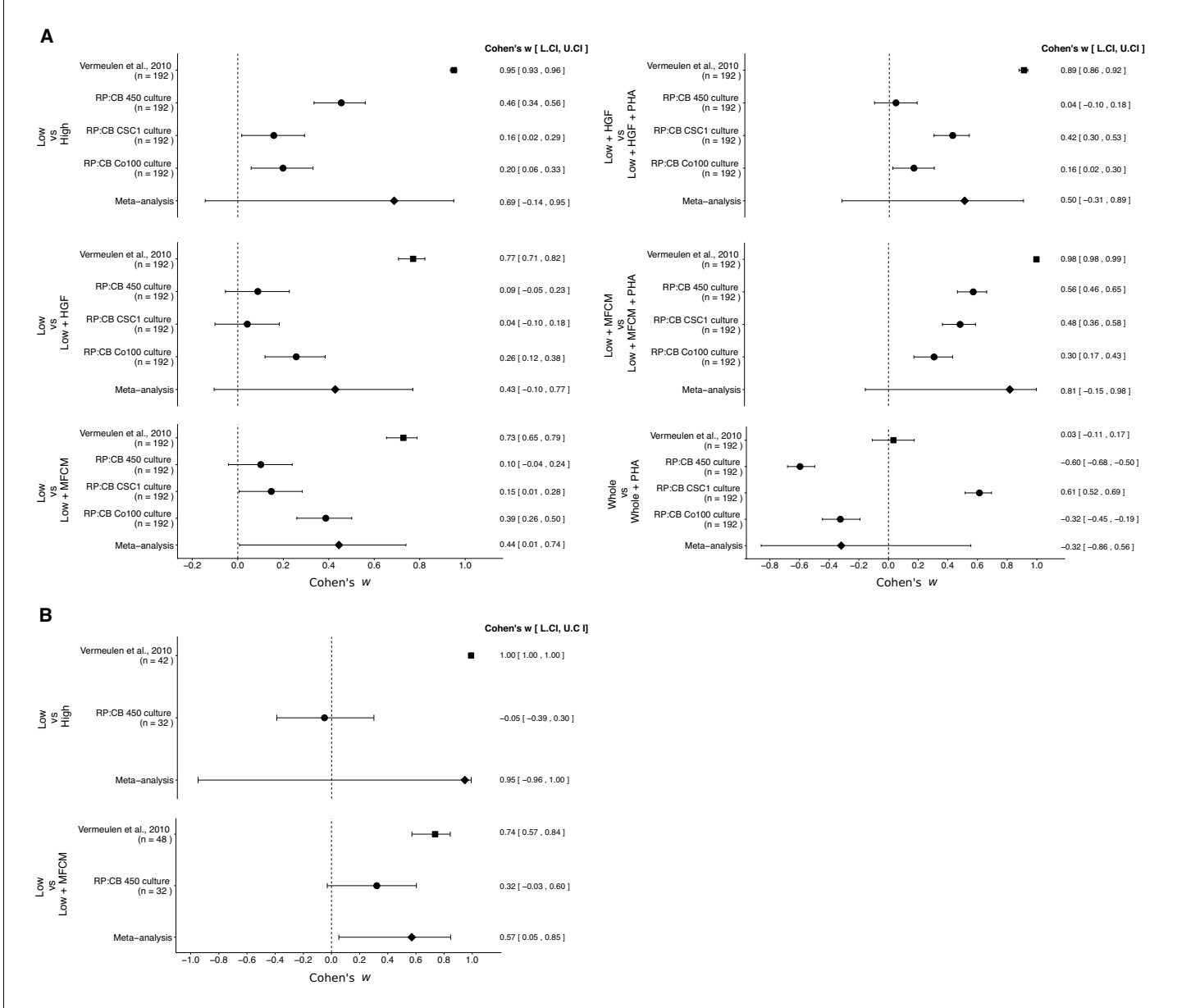

**Figure 3.** Meta-analyses of each effect. Effect size and 95% confidence interval are presented for *Vermeulen et al. (2010)*, the results from this replication study (RP:CB), and a random effects meta-analysis of the effects. Cohen's $\omega$ is a standardized measure of the association between the cells tested and clonogenic, or tumorigenic, frequency. The higher the value, the stronger the association, with an effect size of zero indicating there was no association. Sample sizes used in *Vermeulen et al. (2010)* and RP:CB are reported under the study name. (**A**) Comparison of clonogenic frequency between the indicated treated, or untreated, populations of TOP-GFP CSC cultures. TOP-GFP[low] vs TOP-GFP[high] (meta-analysis $p$=0.094); TOP-GFP[low] vs TOP-GFP[low] + HGF (meta-analysis $p$=0.110); TOP-GFP[low] vs TOP-GFP[low] + MFCM (meta-analysis $p$=0.047); TOP-GFP[low] + HGF vs TOP-GFP[low] + HGF + PHA (meta-analysis $p$=0.218); TOP-GFP[low] + MFCM vs TOP-GFP[low] + MFCM + PHA (meta-analysis $p$=0.085); TOP-GFP[whole] vs TOP-GFP[whole] + PHA (meta-analysis $p$=0.498). (**B**) Comparison of frequency of tumorigenicity between the indicated treated, or untreated, populations of TOP-GFP CSC cultures injected into mice. TOP-GFP[low] vs TOP-GFP[high] (meta-analysis $p$=0.330); TOP-GFP[low] vs TOP-GFP[low] + MFCM (meta-analysis $p$=0.033). Additional details for these meta-analyses can be found at https://osf.io/g4ewk/.
DOI: https://doi.org/10.7554/eLife.45426.007

# Materials and methods

**Key resources table**

| Reagent type (species) or resource | Designation | Source or reference | Identifiers | Additional information |
|---|---|---|---|---|
| Cell line (*Homo sapiens*) | Co100 | DOI: 10.1038/ncb2048 | | shared by Medema lab, University of Amsterdam |
| Cell line (*H. sapiens*, female) | CSC1 | ProMab Biotechnologies | cat# CC100103 | |
| Cell line (*H. sapiens*, female) | E450 | this paper | | |
| Cell line (*H. sapiens*, female) | 18Co | ATCC | cat# CRL-1459; RRID:CVCL_2379 | |
| Strain, strain background (*Mus musculus*, Athymic Nude, female) | athymic nude | Charles River | Strain code: 490; RRID:IMSR_CRL:490 | |
| Recombinant DNA reagent | TOP-GFP | doi: 10.1038/nature01593 | RRID:Addgene_14715 | shared by Medema lab, University of Amsterdam |
| Chemical compound, drug | HGF | Sigma-Aldrich | cat# H5791 | lot# MKBT3102V |
| Chemical compound, drug | PHA-665752 | Sigma-Aldrich | cat# PZ0147 | |
| Other | Matrigel | Corning | cat# 356230 | |
| Antibody | PE-conjugated anti-CD133 | Miltenyi Biotec | cat# 130-098-826; clone: AC133; RRID:AB_2660882 | 1:100 dilution |
| Antibody | PE-conjugated anti-CD24 | BD Biosciences | cat# 560991; clone ML5; RRID:AB_10563074 | 1:100 dilution |
| Antibody | APC-conjugated anti-CD29 | BD Biosciences | cat# 561794; clone: MAR4; RRID:AB_10898163 | 1:100 dilution |
| Antibody | PE-conjugated anti-CD166 | R and D Systems | cat# FAB6561P; clone: 105902; RRID:AB_2223887 | 1:100 dilution |
| Antibody | APC-conjugated anti-CD44 | BD Biosciences | cat# 560890; clone: G44-26; RRID:AB_2033959 | 1:100 dilution |
| Antibody | PE-conjugated mouse IgG1 isotype control | Miltenyi Biotec | cat# 130-098-106; clone: X-56; RRID:AB_2661463 | 1:100 dilution |
| Antibody | APC-conjugated mouse IgG2b, κ isotype control | BD Biosciences | cat# 555745; clone: 27–35; RRID:AB_398612 | 1:100 dilution |
| Antibody | PE-conjugated mouse IgG2a, κ isotype control | BD Biosciences | cat# 555574; clone: G155-178; RRID:AB_395953 | 1:100 dilution |
| Antibody | APC-conjugated mouse IgG1 isotype control | BD Biosciences | cat# 555751; clone: MOPC-21; RRID:AB_398613 | 1:100 dilution |
| Software, algorithm | FACS Sortware sorter | BD Biosciences | RRID:SCR_016722 | version 1.2.0.142 |

*Continued on next page*

*Continued*

| Reagent type (species) or resource | Designation | Source or reference | Identifiers | Additional information |
|---|---|---|---|---|
| Software, algorithm | HCS Studio Cell Analysis | Thermo Fisher Scientific | RRID:SCR_016787 | version 6.6.0 |
| Software, algorithm | FACSDiva | BD Biosciences | RRID:SCR_016722 | version 6.1.3 or 8.0.1 |
| Software, algorithm | FlowJo | Tree Star, Inc | RRID:SCR_008520 | version 10 |
| Software, algorithm | R Project for statistical computing | https://www.r-project.org | RRID:SCR_001905 | version 3.5.1 |

As described in the Registered Report (*Evans et al., 2015*), we attempted a replication of the experiments reported in Figures 2F, 6D, and 7E of *Vermeulen et al. (2010)*. A detailed description of all protocols can be found in the Registered Report (*Evans et al., 2015*) and are described below with additional information not listed in the Registered Report, but needed during experimentation.

## Cell culture

Three cultures of CSCs were isolated/obtained for this study. Co100 cells, which were used in the original study, were shared by Dr. Jan Paul Medema (University of Amsterdam). CSC1 cells were obtained commercially from primary human colorectal tumor tissue from a female Caucasian patient at the age of 65 (ProMab Biotechnologies, cat# CC100103; datasheet available at https://osf.io/det4j/). E450 cells were isolated as described in the Registered Report (*Evans et al., 2015*) from a freshly excised human colon adenocarcinoma tumor fragment from a female Caucasian patient at the age of 78. Of note, E450 was the only viable spheroidal culture that was successfully derived from twelve different colon tissue fragments that were attempted. This is slightly lower (8.3%) then the range of what was shared by the original authors during preparation of the Registered Report (10–20%: *Evans et al., 2015*) and previously reported efficiency rates (15%: *Qureshi-Baig et al., 2016*; 11%: *Brattain et al., 1981*; 33%: *McBain et al., 1984*), although methods to increase efficiency have since been reported (73%; *Miyoshi et al., 2018*). Patient phenotype (e.g. age, sex, ethnicity, diagnosis) for E450 are available at https://osf.io/ysf58/. Approval was obtained from Western Institutional Review Board (WIRB) (protocol MR #0701) and were in full compliance with good clinical practices as defined under the U.S. Food and Drug Administration (FDA) regulations, U.S. Department of Health and Human Services (HHS) regulations, and the International Conference on Harmonisation (ICH) guidelines. Shared samples and data were de-identified for this study. CSCs were maintained at 37°C in a humidified atmosphere at 5% $CO_2$ in CSC medium (modified neurobasal A medium supplemented with 1X N2 supplement, lipid mixture-1 (1 ml/500 ml medium), 20 ng/ml fibroblast growth factor-basic, and 50 ng/ml epidermal growth factor) and passaged as described in the Registered Report (*Evans et al., 2015*) with additional details available at https://osf.io/dtbvp/. 18Co cells (ATCC, cat# CRL-1459, RRID:CVCL_2379) were maintained at 37°C in a humidified atmosphere at 5% $CO_2$ in Eagle's Minimum Essential Medium supplemented with 10% FBS, 500 U/ml penicillin, 500 U/ml streptomycin, and 1.25 µg/ml amphotericin B. Quality control data confirming the cells were free of mycoplasma contamination (MycoAlert Mycoplasma Detection kit; Lonza, cat# LT07-318) is available at https://osf.io/xzh9t/.

## Lentiviral infection

TCF/LEF-1 responsive promoter to drive expression of green fluorescent protein (GFP) (TOP-GFP, RRID:Addgene_14715) was shared Dr. Jan Paul Medema (University of Amsterdam) with permission from Dr. Laurie Ailles (University Health Network; University of Toronto). Spheroidal cultures were transduced with lentiviral particles to express TOP-GFP which were produced by Cyagen Biosciences, Inc (Santa Clara, California) with a titer of $3.66 \times 10^8$ TU/ml as determined by quantitative PCR using a fragment in the WPRE region of the lentiviral vector amplified from genomic DNA of transduced HEK293 cells. Dissociated spheroidal cultures were each transduced for 24 hr with 20 µl concentrated lentivirus per $1 \times 10^6$ cells in 10 ml CSC medium supplemented with 8 µg/l polybrene before medium was replaced. Cells were cultured for 4 weeks before isolation of single-cell-derived

cultures by fluorescence-activated cell sorting (FACS). FACS was performed on an Influx cell sorter (BD Biosciences) and analyzed with FACS Sortware sorter software (BD Biosciences, RRID:SCR_016722), version 1.2.0.142. Spheroids were dissociated as described in the Registered Report (*Evans et al., 2015*) and propidium iodide (PI) was added at 250 ng/ml immediately prior to sorting. Single, PI-negative, GFP-positive cells were sorted and deposited into individual wells of ultralow-adhesion 96-well plates containing 200 µl/well CSC medium. Four 96-well plates were tested for each culture with one viable single-cell clone generated from the CSC1 culture (0.26% efficiency), three clones from the E450 culture (0.78% efficiency, and eight clones from the Co100 culture (2.08% efficiency), which were near the range of what was shared by the original authors during preparation of the Registered Report (~1%: *Evans et al., 2015*). One clone was randomly selected from each culture for further analysis. Over a period of 13 weeks, spheroid cultures arising from single cells were gradually expanded into larger ultralow-adhesion flasks. Microscopy images of cultures were acquired with a CellInsight CX7 High-Content Screening (HCS) Platform (ThermoFisher Scientific) and HCS Studio Cell Analysis software (ThermoFisher Scientific, RRID:SCR_016787) version 6.6.0, build 8153.

## Flow cytometry analysis of cell surface markers

Spheroid cultures were dissociated with trypsin and resuspended at a final concentration of $1 \times 10^6$ cells/ml in FACS buffer (PBS supplemented with 2% fetal bovine serum (FBS), 1X antifungal/antibiotic, and 2 mM EDTA). Cells were stained at 1:100 dilution with PE-conjugated monoclonal anti-CD133 (Miltenyi Biotec, cat# 130-098-826, clone AC133, RRID:AB_2660882), PE-conjugated monoclonal anti-CD24 (BD Biosciences, cat# 560991, clone ML5, RRID:AB_10563074) and APC-conjugated monoclonal anti-CD29 (BD Biosciences, cat# 561794, clone MAR4, RRID:AB_10898163), or PE-conjugated monoclonal anti-CD166 (R and D Systems, cat# FAB6561P, clone 105902, RRID:AB_2223887) and APC-conjugated monoclonal anti-CD44 (BD Biosciences, cat# 560890, clone G44-26, RRID:AB_2033959) and incubated at 4°C in the dark for 10 min. Cells were also stained with 1:100 dilution of control antibodies: PE-conjugated monoclonal mouse IgG1 isotype control (Miltenyi Biotec, cat# 130-098-106, clone X-56, RRID:AB_2661463), APC-conjugated monoclonal mouse IgG2b, κ isotype control (BD Biosciences, cat# 555745, clone 27–35, RRID:AB_398612), PE conjugated monoclonal mouse IgG2a, κ isotype control (BD Biosciences, cat# 555574, clone G155-178, RRID:AB_395953), or APC-conjugated monoclonal mouse IgG1 isotype control (BD Biosciences, cat# 555751, clone MOPC-21, RRID:AB_398613). Cells were washed by adding 20 times the reaction volume of FACS buffer and gently inverting tubes three times. Cells were centrifuged at 1000 RPM for 10 min, supernatant was carefully aspirated, and the cells were resuspended in 100 µl FACS buffer. Flow cytometry analysis was performed on a FACSAria II (BD Biosciences) and analyzed with FACSDiva software (BD Biosciences, RRID:SCR_016722), version 6.1.3. PI (250 ng/ml) was added to cells just before analysis. FACS data was imported into FlowJo software (Tree Star, Inc, RRID:SCR_008520), version 10, after which the scaled compensated values were exported as csv files. These values were then imported into Python 2.7 to perform rectangular gating. Cells were first gated using the forward scatter and propidium-iodide channels (i.e. cells negative for propidium-iodide were retained). Cells were subsequently gated for positive TOP-GFP expression. After this, cells below the 10th percentile and above the 90th percentile of TOP-GFP expression were compared. Gating strategy was described in the Registered Report with additional details available at https://osf.io/8c43g/ and a representative example depicted in *Figure 1—figure supplement 1*.

## Conditioned medium

$7.5 \times 10^5$ 18 Co cells were seeded in a 75 cm$^2$ flask and incubated overnight. The next day, cells were washed twice with PBS and incubated for 24 hr with 10 ml of CSC medium without EGF and FGF-basic. The next day the conditioned medium was collected and cleared by centrifugation for 5 min at 1400 RPM and used at 1:2 dilution in CSC medium for the assays described below. The level of HGF present in MFCM was determined by ELISA (Sigma-Aldrich, cat# RAB0212) according to manufacturer's instructions with a standard curve. Data are available at https://osf.io/fpj4u/.

## Limiting-dilution assay

An initial pilot experiment was performed to assess the potential for the three populations (TOP-GFP$^{low}$ (10% lowest), TOP-GFP$^{high}$ (10% highest), and TOP-GFP$^{whole}$ (total)), without treatment, on the three TOP-GFP CSC cultures (Co100, CSC1, E450). Cells from the different populations were deposited at 1, 2, 4, 6, 8, 12, 16, 20, and 24 cells per well in the number of wells indicated in the Registered Report with additional details available at https://osf.io/ydfrg/. Cells were deposited with an Influx cell sorter and analyzed with FACS Sortware sorter software, version 1.2.0.142. Cells were incubated at 37°C in a humidified atmosphere at 5% $CO_2$, with culture medium replaced every 4 days. After 14 days of culture, the number of cultures with spheres, and the number of cells per sphere were quantified using automated high-content fluorescence imaging for GFP-positive and Hoechst-positive cells using a CellInsight CX7 High-Content Screening (HCS) Platform and HCS Studio Cell Analysis software. Spheres composed of two or more cells were used to determine clonal frequency which was evaluated by ELDA from the *statmod* R package (*Hu and Smyth, 2009*), version 1.4.30. Raw data are available at https://osf.io/ctqu2/ with data aggregated in csv format (https://osf.io/ydejb/). Pilot results reported in *Figure 2—figure supplement 1*. Based on these results, it was decided that the cell titration would remain the same for the confirmatory experiment, and that the E450 culture would be used for the *in vivo* tumorigenicity assay. Cells from the indicated TOP-GFP population were deposited into 96-well ultralow-adhesion plates with 100 µl of either CSC medium (untreated), CSC medium with 25 ng/ml HGF (Sigma-Aldrich, cat# H5791, lot# MKBT3102V), CSC medium with MFCM (1:2 dilution in CSC medium), CSC medium with 25 ng/ml HGF and 500 nM PHA-665752 (Sigma-Aldrich, cat# PZ0147), CSC medium with MFCM and 500 nM PHA-665752, or CSC medium with 500 nM PHA-665752. Cells were deposited with a FACSAria III (BD Biosciences) and analyzed with FACSDiva software, version 8.0.1. Cells were incubated at 37°C in a humidified atmosphere at 5% $CO_2$, with the appropriate culture medium replaced every 4 days. After 14 days of culture, the number of cultures with spheres, and the number of cells per sphere were blindly quantified using automated high-content fluorescence imaging for GFP-positive and Hoechst-positive cells as described for the pilot assay. Raw data are available at https://osf.io/qwgx4/ with data aggregated in csv format (https://osf.io/26zp5/). Clonal frequency and statistical significance was determined by ELDA (*Hu and Smyth, 2009*).

## *In vivo* tumorigenicity assay

All animal procedures were approved by the Explora BioLabs, Inc animal use committee (IACUC# SP17-009-005A) and were in accordance with Explora BioLabs, Inc policies on the care, welfare, and treatment of laboratory animals.

Nine-week old female Athymic Nude mice (Charles River, Strain code: 490, RRID:IMSR_CRL:490) were housed in sterile conditions under standard temperature, humidity, and timed lighting conditions with 12 hr light/dark cycles and acclimated to the housing environment for 3 days prior to the initiation of the study. They were housed on bedding material (Corn Cobb, cat# M-BTM-C8) that was changed bi-weekly. Animals were provided standard diet (Envigo, cat# 2920X (Irradiated Global 18% Soy Protein Extruded Rodent Diet)) and acidified water (pH 2.5–3.0) throughout the study period *ad libitum*. Body weights were measured on Day −1 for randomization and the 48 animals were stratified into 12 groups to obtain similar average body weight among groups. Following cell-injection, animal health, body weight, and tumor observation were recorded weekly and are available at https://osf.io/xs9up/. Tumor volumes were calculated from caliper measurements using the formula (volume = 1/2(length*width$^2$). Experimental work was performed blinded to the identity of the sample the mice were injected with.

Mice were injected, on Day 0, with TOP-GFP transduced cultures (E450 culture) at 10, 100, 1000, or 5000 cells from the 10% lowest or 10% highest TOP-GFP intensities that were deposited, by FACS (FACSAria II with FACSDiva software, version 6.1.3), in a 96-well ultralow-adhesion plate and resuspended in 100 µl of CSC medium or MFCM (generated as described above and in the Registered Report [*Evans et al., 2015*]) and incubated at 37°C for 2 hr. After this incubation, plates were shipped to the facility that performed the mouse injection/monitoring (~30 min) where the cells and medium (100 µl) were mixed with growth factor reduced Matrigel (Corning, cat# 356230) at a 1:1 ratio and injected subcutaneously into the right flank of the female mice using a sterile 25 G needle and 1 ml syringe as described in the Registered Report (*Evans et al., 2015*). Mice were monitored

for tumor formation for nine weeks after injection, the indicated study endpoint in the Registered Report. To explore if the frequency changed, the mice were monitored an additional 2 weeks; however we did not observe any new tumor initiations. Tumor-initiating cell frequency was determined by ELDA (*Hu and Smyth, 2009*).

## Statistical analysis

Statistical analysis was performed with R software (RRID:SCR_001905), version 3.5.1 (*R Development Core Team, 2018*). All data, csv files, and analysis scripts are available on the OSF (https://osf.io/pgjhx/). Confirmatory statistical analysis was pre-registered (https://osf.io/rscpj/) before the experimental work began as outlined in the Registered Report (*Evans et al., 2015*). Data were checked to ensure assumptions of statistical tests were met. The fitted models to determine the stem cell frequency for different groups were compared using likelihood ratio tests using the asymptotic chi-square approximation to the log-ratio (*Hu and Smyth, 2009*). When described in the results, the Bonferroni correction, to account for multiple testings, was applied to the alpha error or the *p*-value. The Bonferroni corrected value was determined by divided the uncorrected value (0.05) by the number of tests performed. The confidence intervals for the Cohen's $\omega$ estimates were determined using a Fisher's z' transformation (*Rosenthal and DiMatteo, 2001*). A meta-analysis of a common original and replication effect size was performed with a random effects model and the *metafor* R package (*Viechtbauer, 2010*), version 2.0–0 (https://osf.io/g4ewk/). The original study data of the stem cell frequency and 95% CI pertaining to Figure 6D was extracted *a priori* from the published figure and used to create simulated data sets with preserved sampling structure using ELDA (*Hu and Smyth, 2009*) during preparation of the experimental design, while the original study data pertaining to Figure 7E was published in the original paper (*Vermeulen et al., 2010*). The C100.B5 line from the original study was used in the meta-analysis for the tumorigenicity assay, but not the C100.G7 line because an estimate for TOP-GFP$^{low}$ could not be calculated (i.e. estimate was infinity because of no observable responses). The summary data was published in the Registered Report (*Evans et al., 2015*) and used in the power calculations to determine the sample sizes for this study.

## Data availability

Additional detailed experimental notes, data, and analysis are available on OSF (RRID:SCR_003238) (https://osf.io/pgjhx/; *Essex et al., 2019*). This includes the R Markdown file (https://osf.io/d6qp8/) that was used to compose this manuscript, which is a reproducible document linking the results in the article directly to the data and code that produced them (*Hartgerink, 2017*). Flow cytometry data for this study has also been deposited at Flow Repository (RRID:SCR_013779; *Spidlen et al., 2012*), where it is directly accessible at https://flowrepository.org/id/FR-FCM-ZYUG.

## Deviations from registered report

We planned to isolate two CSC spheroidal cultures from patient samples, but due to only obtaining a single viable spheroidal culture from the 12 different colon tissue fragments that were attempted, we obtained another culture commercially that was also derived from primary human colorectal tumor tissue. The Registered Report indicated we would perform the flow cytometry analysis and clonogenicity assay on three different single-cell TOP-GFP clones from each of the three cultures, while the results reported in this study are from one random single-cell TOP-GFP clone from each culture. This was due to the CSC1 culture only producing one viable clone as stated in the 'Lentiviral infection' section above. We also did not perform the statistical analysis listed in Protocol 2 for the cell surface markers since the observed variation was from the same population of cells. Additional materials and instrumentation not listed in the Registered Report, but needed during experimentation are also listed.

## Acknowledgements

The Reproducibility Project: Cancer Biology would like to thank Dr Jan Paul Medema (University of Amsterdam) and Dr Giorgio Stassi (University of Palermo) for sharing critical information, data, and reagents, specifically the Co100 cells and TOP-GFP plasmid used in the original study. We want to thank Dr Laurie Ailles (University Health Network; University of Toronto) for providing permission for the TOP-GFP plasmid. We want to thank I Janette Delgadillo (Explora BioLabs, Inc) for providing

study oversight for the animal experiments and Dr Lawrence R Blocher (Minerva Resource, Inc) for human colon adenocarcinoma tumor fragment collection. We would also like to thank the following companies for generously donating reagents to the Reproducibility Project: Cancer Biology; American Type and Tissue Collection (ATCC), Applied Biological Materials, BioLegend, Charles River Laboratories, Corning Incorporated, DDC Medical, EMD Millipore, Harlan Laboratories, LI-COR Biosciences, Mirus Bio, Novus Biologicals, Sigma-Aldrich, and System Biosciences (SBI).

## Additional information

### Group author details

**Reproducibility Project: Cancer Biology**
**Elizabeth Iorns**: Science Exchange, Palo Alto, United States; **Rachel Tsui**: Science Exchange, Palo Alto, United States; **Alexandria Denis**: Center for Open Science, Charlottesville, United States; **Nicole Perfito**: Science Exchange, Palo Alto, United States; **Timothy M Errington**: Center for Open Science, Charlottesville, United States; **Elizabeth Iorns**: Science Exchange, Palo Alto, United States; **Rachel Tsui**: Science Exchange, Palo Alto, United States; **Alexandria Denis**: Center for Open Science, Charlottesville, United States; **Nicole Perfito**: Science Exchange, Palo Alto, United States; **Timothy M Errington**: Center for Open Science, Charlottesville, United States

### Competing interests

Anthony Essex, Javier Pineda, James Evans: PhenoVista Biosciences is a Science Exchange associated lab. Reproducibility Project: Cancer Biology: EI, RT, NP: Employed by and hold shares in Science Exchange Inc. The other authors declare that no competing interests exist.

### Funding

The Reproducibility Project: Cancer Biology is funded by the Laura and John Arnold Foundation, provided to the Center for Open Science in collaboration with Science Exchange. The funder had no role in study design, data collection and interpretation, or the decision to submit the work for publication.

### Author contributions

Anthony Essex, Javier Pineda, James Evans, Acquisition of data, Analysis and interpretation of data, Drafting or revising the article, Conducted in vitro experimentation; Grishma Acharya, Acquisition of data, Analysis and interpretation of data, Drafting or revising the article, Conducted in vivo experimentation; Hong Xin, Reproducibility Project: Cancer Biology, Analysis and interpretation of data, Drafting or revising the article

### Author ORCIDs

Alexandria Denis (iD) https://orcid.org/0000-0002-1210-2309
Timothy M Errington (iD) https://orcid.org/0000-0002-4959-5143
Alexandria Denis (iD) https://orcid.org/0000-0002-1210-2309
Timothy M Errington (iD) https://orcid.org/0000-0002-4959-5143

### Ethics

Human subjects: Approval was obtained from Western Institutional Review Board (WIRB) (protocol MR #0701) and were in full compliance with good clinical practices as defined under the US Food and Drug Administration (FDA) regulations, US Department of Health and Human Services (HHS) regulations, and the International Conference on Harmonisation (ICH) guidelines.
Animal experimentation: All animal procedures were approved by the Explora BioLabs, Inc animal use committee (IACUC# SP17-009-005A) and were in accordance with Explora BioLabs, Inc policies on the care, welfare, and treatment of laboratory animals.

**Decision letter and Author response**
Decision letter https://doi.org/10.7554/eLife.45426.016
Author response https://doi.org/10.7554/eLife.45426.017

## Additional files

### Supplementary files

• Transparent reporting form
DOI: https://doi.org/10.7554/eLife.45426.008

• Reporting standard 1. The ARRIVE guidelines checklist.
DOI: https://doi.org/10.7554/eLife.45426.009

### Data availability

Additional detailed experimental notes, data, and analysis are available on OSF (RRID:SCR_003238) (https://osf.io/pgjhx/; Essex et al., 2019). This includes the R Markdown file (https://osf.io/d6qp8/) that was used to compose this manuscript, which is a reproducible document linking the results in the article directly to the data and code that produced them (Hartgerink, 2017). Flow cytometry data for this study has also been deposited at Flow Repository (RRID:SCR_013779; Spidlen et al., 2012), where it is directly accessible at https://flowrepository.org/id/FR-FCM-ZYUG.

The following datasets were generated:

| Author(s) | Year | Dataset title | Dataset URL | Database and Identifier |
|---|---|---|---|---|
| Essex A, Pineda J, Acharya G, Xin H, Evans J, Iorns E, Tsui R, Denis A, Perfito N, Errington TM | 2019 | Study 9: Replication of Vermeulen et al., 2010 (Nature Cell Biology) | http://dx.doi.org/10.17605/OSF.IO/PGJHX | Open Science Framework, 10.17605/OSF.IO/PGJHX |
| Essex A, Pineda J, Acharya G, Xin H, Evans J, Iorns E, Tsui R, Denis A, Perfito N, Errington TM | 2019 | Replication Study: Wnt activity defines colon cancer stem cells and is regulated by the microenvironment | https://flowrepository.org/id/FR-FCM-ZYUG | FlowRepository, FR-FCM-ZYUG |

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
