## [Decision Letter]

Thank you for submitting your work entitled "Replication Study: Wnt activity defines colon cancer stem cells and is regulated by the microenvironment" for consideration at *eLife*. Your article has been evaluated by Sean Morrison as the Senior Editor, a Reviewing Editor, and three reviewers.

There are a few issues identified by the reviewers that need to be addressed before acceptance, as outlined below:

1) Address the concerns raised by reviewer 1 regarding the limiting dilution data as well as provide the primary data for these experiments. In addition, specify the type of Matrigel used in the in vivo experiments given the divergence in results between the two studies.

2) As indicated by reviewer 2, comment on any possible discrepancy between HGF production in conditioned medium from the original study to this replication study that may account for variation in results. Also, comment on subsequent work that has linked WNT signatures to clonogenicity/stemness and if any have validated these particular markers in CRC cells.

*Reviewer #1:*

The current Replication Study indicates that the original observations can be reproduced to a large extent. Several key features show clear differences from the original study. These include clonogenic efficacy, expression of specific markers and in vivo tumorigenicity. Below a point by point assessment of this differences is given as well as several other details.

1) In the Registered Report it was noted that multiple subclones would be tested. It is unclear why the authors decided to divert from this original plan and only test one clone. This should be explained, but maybe more important, be revised.

2) The in vivo tumorigenicity is the main feature that is different from the Original study. In the report it is not noted what Matrigel was used to inject the tumor cells. If this is not growth factor reduced the data can be explained quite easily. This information should be given.

3) CD24 and in particular CD44, are well-established Wnt target genes and reporters (e.g. Wielenga et al., 1999). The current authors report that Wnt-low cells display lower CD44 and CD24 levels in some lines. This warrants further discussion.

4) Widespread variation in clonogenic potential with specific cell populations has been reported beyond Wnt activity levels. For example Cd133+ cells are (Ricci-Vitiani et al., 2007, and O'Brien et al., 2007) or are not (Shmelkov et al., 2008) associated with the tumorigenic population. In addition, even in similar lines different results are obtained; for example compare Dittfeld et al., 2009 with Chen et al., 2011 regarding the use of Cd133 in HCT116 cells to define a clonogenic subfraction. The current results should be described in this context.

5) The establishment of lines from primary tissue is with 1 in 12 extremely inefficient in comparison with our experience (~50%) and published literature (50-90%). This should be described better as it may impact on the data as well.

*Reviewer #2:*

I do not have major technical concerns for this replication study that would require additional experimental work.

*Reviewer #3:*

Essex, Evans and colleagues have undertaken a replication analysis of selected studies from the 2010 manuscript by Vermeulen et al. (Vermeulen et al., 2010). The replication study focused on cell surface markers of presumptive Wnt pathway-high cancer stem cells, the clonogenicity in vitro of presumptive Wnt pathway-high cancer stem cells vs. Wnt pathway-low cells and how cancer-associated myofibroblast-associated secreted factors might influence clonogenicity; and the tumorigenic growth potential of Wnt pathway-high cells and how tumor-associated myofibroblast-secreted factors might impact on tumorigenic potential.

The replication study was able to show that only three of the five cell surface markers previously linked to Wnt pathway-high transcriptional activity status in three spheroid cancer cell line models could be confirmed. The authors confirmed that in vitro clonogenicity was highest in cells with Wnt pathway-high status and that clonogenic potential of cells low Wnt pathway activity was increased by myofibroblast-secreted factors, such as HGF. Some of the findings on clonogenicity were of less certain statistical significance. The authors did not find a difference in tumorigenicity between the cells with Wnt pathway-high vs. Wnt pathway-low transcriptional status, though tumorigenicity was increased with myofibroblast-secreted factors, but not in a statistically significant manner.

The replication work is solidly performed and presented and interpreted in a reasonable manner.

No major comments are offered.

---

## [Author Response]

1) Address the concerns raised by reviewer 1 regarding the limiting dilution data as well as provide the primary data for these experiments. In addition, specify the type of Matrigel used in the in vivo experiments given the divergence in results between the two studies.

The limiting dilution data were available via a private link, which will later be made public. We have also added the data as figure 2—figure supplement 1. In the revised manuscript we also included in the source of the Matrigel used, which was growth factor reduced as described in the Registered Report.

2) As indicated by reviewer 2, comment on any possible discrepancy between HGF production in conditioned medium from the original study to this replication study that may account for variation in results. Also, comment on subsequent work that has linked WNT signatures to clonogenicity/stemness and if any have validated these particular markers in CRC cells.

We have included further discussion on these two aspects in the revised manuscript.

Reviewer #1:

*The current Replication Study indicates that the original observations can be reproduced to a large extent. Several key features show clear differences from the original study. These include clonogenic efficacy, expression of specific markers and* in vivo tumorigenicity. Below a point by point assessment of this differences is given as well as several other details.1) In the Registered Report it was noted that multiple subclones would be tested. It is unclear why the authors decided to divert from this original plan and only test one clone. This should be explained, but maybe more important, be revised.

In the revised article we have further explained this deviation in the ‘Lentiviral infection’ and ‘Deviations from Registered Report’ sections of the Materials and methods.

*2) The* in vivo *tumorigenicity is the main feature that is different from the Original study. In the report it is not noted what Matrigel was used to inject the tumor cells. If this is not growth factor reduced the data can be explained quite easily. This information should be given.*

The Matrigel used was growth factor reduced (Corning, cat# 356230), as described in the Registered Report. We have also included this information in the revised manuscript.

3) CD24 and in particular CD44, are well-established Wnt target genes and reporters (e.g. Wielenga et al., 1999). The current authors report that Wnt-low cells display lower CD44 and CD24 levels in some lines. This warrants further discussion.

We have added further discussion on this observation in the revised manuscript.

4) Widespread variation in clonogenic potential with specific cell populations has been reported beyond Wnt activity levels. For example Cd133+ cells are (Ricci-Vitiani et al., 2007, and O'Brien et al., 2007) or are not (Shmelkov et al., 2008) associated with the tumorigenic population. In addition, even in similar lines different results are obtained; for example compare Dittfeld et al., 2009 with Chen et al., 2011 regarding the use of Cd133 in HCT116 cells to define a clonogenic subfraction. The current results should be described in this context.

We have added this additional context in the revised manuscript.

5) The establishment of lines from primary tissue is with 1 in 12 extremely inefficient in comparison with our experience (~50%) and published literature (50-90%). This should be described better as it may impact on the data as well.

We have added additional references on the efficiency of establishing lines from primary tissue in the revised manuscript, which are in the range of 11-33%, as well as additional methods that have been published recently demonstrating much higher efficiency (73%). Additionally, of note, in correspondence with the original authors, we were informed that the success rate was 10-20% to form spheroidal cultures from primary tissue samples, which is close to what we observed.